

# Sequence analysis and mRNA expression of prolactin receptor gene isoforms in different tissues of sheep during lactation and the post-weaning period

Ruochen Yang[1,*], Chunhui Duan[1,*], Yunxia Guo[2], Yujing Ma[1], Nazi Niu[2], Yingjie Zhang[1] and Yueqin Liu[1]

[1] College of Animal Science and Technology, Hebei Agricultural University, Baoding, Hebei, China
[2] College of Life Sciences, Hebei Agricultural University, Baoding, Hebei, China
[*] These authors contributed equally to this work.

Corresponding authors
Yingjie Zhang,
zhangyingjie66@126.com
Yueqin Liu, liuyueqin66@126.com

## ABSTRACT

Few studies on mRNA expression of the prolactin receptor (*PRLR*) isoforms in different tissues of sheep were reported. The objective of this study was to analyze the gene sequence and mRNA expression of *PRLR* isoforms in the uterus, mammary gland, ovary, spleen and lymph tissue of ewes during the lactation and post-weaning periods. Ten lactating crossbred ewes (Dorper×Hu sheep) with twin lambs were used in this study. Five ewes were chosen randomly and slaughtered at mid-lactation (35 days after lambing). The remaining five ewes were slaughtered on the 5th day after weaning. Samples of uterus, mammary gland, ovary, spleen and lymph tissue were collected from each ewe to determine the mRNA expression of long *PRLR* (*L-PRLR*) and short *PRLR* (*S-PRLR*) by RT-qPCR. The physical and chemical properties, the similarity of the nucleotides *L-PRLR* and *S-PRLR* genes and the secondary and tertiary structure of the *L-PRLR* and *S-PRLR* proteins of sheep were analyzed. The results indicated that the predicted protein molecular weights of *L-PRLR* and *S-PRLR* are 65235.36 KD and 33847.48 KD, respectively, with isoelectric points of 5.12 and 8.34, respectively. The secondary protein structures of *L-PRLR* and *S-PRLR* are different. For *L-PRLR* these include alpha helix, extended strand and random coils and β-turns for which the content was 16.01%, 21%, 59.55% and 3.44%, respectively, whereas the secondary protein structures of *S-PRLR* contain only alpha helices, extended strand and random coils, comprising 18.24%, 30.07% and 48.99%, respectively. The *L-PRLR* and *S-PRLR* genes of the sheep (*Ovis aries*) had nucleotide sequences showing much similarity among ruminants. In these sheep, mRNA expression of *L-PRLR* and *S-PRLR* was highest in the uterus and differed between the uterus, ovary, mammary gland, spleen and lymph tissue. The mRNA expression of *L-PRLR* in lymph tissue was higher during lactation than in the post-weaning period ($P < 0.01$), whereas mRNA expression of *S-PRLR* in the uterus and the mammary gland was lower during lactation than during the post-weaning period ($P < 0.01$). In the uterus, mRNA expression of *L-PRLR* was higher than that of *S-PRLR* during lactation ($P < 0.01$) but there were no significant differences ($P < 0.05$) for the other five tissues. This study that the *L-PRLR* and *S-PRLR* proteins in ewes are mainly composed of extended fragments and random coils. The data also indicate that mRNA expression of *L-PRLR* and *S-PRLR* genes varies among different

tissues in sheep and is higher in the uterus than in the ovary, spleen, mammary gland and lymph tissue throughout lactation and the post-weaning period.

## INTRODUCTION

Prolactin (*PRL*) is a single chain polypeptide hormone synthesized and secreted from the anterior pituitary gland. It belongs to the prolactin/growth hormone family (*Goffin et al., 1996*) and participates in various physiological processes in mammals such as reproduction, immunity and regulation of metabolism (*Freeman et al., 2000*). The prolactin receptor, *PRLR,* has a central role in the *PRL* signal transduction cascade since *PRL* exerts its biological functions by binding to *PRLR* (*Bignon et al., 1997*). *PRLR* belongs to the superfamily of cytokine receptors and has been detected in a variety of tissues in many mammals (*Motamedi et al., 2020*). Ruminants have both a long and a short prolactin receptor (*L-PRLR*, *S-PRLR,* respectively) (*Bignon et al., 1997*). Their genes, *L-PRLR* and *S-PRLR,* are expressed by alternative splicing of a single *PRLR* gene (*Chen et al., 2020*) and they differ by the lengths of their carboxyl-terminals at the cytoplasmic domains (*Viitala et al., 2006*).

Many previous studies have reported that *PRLR* is associated with reproduction (*Goffin et al., 1998*). In rodent ovaries mRNA expression of *L-PRLR* is higher than that of *S-PRLR* during all phases of the estrous cycle and throughout pregnancy (*Clarke, Arey & Linzer, 1993*; *Clarke & Linzer, 1993*). *PRLR* has also been reported to be associated with immunity (*Zhou et al., 2020*). For instance, the circulating concentration of PRL directly affects the production of $CD_S^+T$ cells (*Bernichtein, Touraine & Goffin, 2010*) and the role of PRL needs to be achieved by the expression of *PRLR* on immune cells (*Zhou et al., 2020*). *S-PRLR* has been cloned in rats (*Boutin et al., 1988*), and *L-PRLR* is widely expressed in the muscle, liver, spleen, mammary gland and adipose tissues of dairy goats (*Shi et al., 2016*). However, few studies on mRNA expression of the *PRLR* isoforms in different tissues of sheep were reported. We hypothesized that the expression of *L-PRLR* and S-PRLR were different in uterus, ovary, mammary gland, spleen and lymph tissue of sheep during the different physiological phases. If it does, it will be of great value for further research on the function of *L-PRLR* and *S-PRLR* in ruminants. Therefore, the objective of the present study was to analyze the gene sequence of *PRLR* isoforms and the mRNA expression of *L-PRLR* and *S-PRLR* in the ovary, mammary gland, uterus, lymph tissue and spleen in ewes of the Dorper ×Hu breed during the lactation and post-weaning periods.

## MATERIALS AND METHODS

### Animals and experimental design

The study was conducted from August to October 2018 on Weizun Sheep Farm located in the Hebei province of China. All procedures used in this study were approved by the

Laboratory Animal Ethics Committee of Hebei Agricultural University (Hebei, P.R. China; permit number 2018082).

Total mixed rations (TMR) were formulated according to NRC (2007). Ewes were fed twice daily, at 0700 h and 1700 h, and had free access to clean water. Feed residuals were 5%–7% of the total offered and these were removed when cleaning was carried out each day after the afternoon feeding.

Ten lactating crossbred ewes (Dorper × Hu sheep, 2.5 years of age) with twin lambs were used. Five ewes were chosen randomly and euthanized in the middle (35 days after lambing) of the lactation period. The remaining five ewes were euthanized on the 5th day after weaning. Samples of tissue from the uterus, mammary gland, ovary, spleen and lymph tissue were collected from each ewe after euthanized.

## Data and sample collection

Ten ewes were deprived of feed for 24 h and of water for 16 h then killed humanely at around 0800 h using electro-stunning followed by severance of blood vessels in the neck. Samples of tissue from the uterus, mammary gland, ovary, spleen and lymph tissue were obtained within 20 min of death under sterile conditions by using a sterile scalpel. These were cut into 0.2 cm$^3$ pieces and then immediately frozen in liquid nitrogen for storage until extraction of RNA for the analysis of *PRLR* expression.

## Analytical procedures
### Primer design
Conserved regions were found by aligning sheep *L-PRLR*, *S-PRLR* and *GADPH* gene sequences published in GenBank using DNAMAN. Primers were then designed from the conserved region using Primer Premier 5.0 and synthesised by Shanghai Sheng Gong Biotechnology Co., Ltd. Primer sequences and related information are as follows: L-PRLR: 5′-CCCCTTGTTCTCTGCTAAACCC-3′(forward), 5′-CTATCCGTCACCCGAGACACC-3′(reverse) (120 bp); S-PRLR: 5′-AAATACCTTGTGCAGATTCGATG-3′(forward), 5′-AAACACAGACACAAGGCGAGA-3′(reverse) (267 bp); GAPDH: 5′-CTGACCTGCCGCCTGGAGAAA-3′(forward), 5′-GTAGAAGAGTGAGTGTCGCTGTT-3′(reverse) (149 bp).

## RNA extraction and quantitative real-time PCR
After thoroughly grinding the collected tissues in liquid nitrogen, total RNA was extracted according to the specification of TRNzol total RNA extraction kit (TIANGEN) and stored at −80 °C. Reverse transcription was carried out using a reagent kit (Takala) according to the manufacturer's instructions. The reaction mixture, 20 μL, consisted of 5×Mix (4 μL), RNA (2 μg), and RNase-free water (16 μL). The reaction was performed at 37 °C for 15 min followed by 85 °C for 5 s; the product was stored at −4 °C.

q-PCR was conducted in strict accordance with the LC-480 PCR system instructions using Ultra SYBR Mixture (with Rox) and the following cycling protocol: 10 min at 95 °C followed by 40 cycles consisting of 15 s at 95 °C and 60 s at 60 °C. The reaction mixtures contained 10 μL Ultra SYBR Mixture (2×), 0.4 μL upstream and downstream primers (10 μM) each, 2 μL template and sterile distilled water to a final volume of 20 μL.
**Table 1** The nucleotide sequence similarity of the CDS region of *L-PRLR* and *S-PRLR* (%).

| Species | Capra hircus | Bos taurus | Sus scrofa | Mus musculus | Homo sapiens | Gallus gallus |
|---------|--------------|------------|------------|--------------|--------------|---------------|
| *L-PRLR* | 97.65 | 94.79 | 77.16 | 65.12 | 71.86 | 39.24 |
| *S-PRLR* | 98.88 | 96.18 | 85.20 | 16.70 | 65.16 | 53.70 |

Relative expression of the target genes was calculated by the $2^{-\Delta\Delta CT}$ method based on the quantitative real-time PCR results.

### Sequencing analysis

EditSeq software was used to predict the physical and chemical properties of *L-PRLR* and *S-PRLR* in ewes, which included amino acid composition, isoelectric point and theoretical molecular weight; DNAMAN software was used to analyze the similarity of the nucleotides of *L-PRLR and S-PRLR*. Online software was used to predict the secondary structure of *L-PRLR* and *S-PRLR* proteins (https://npsa-prabi.ibcp.fr/cgi-bin/npsa_automat.pl?page=npsa_sopma.html) and their tertiary structure (http://www.expasy.org/swissmod/swiss-model.html).

### Statistical analysis

Data were analyzed by one-way analysis of variance (ANOVA) and *T*-test for comparisons (*T*-test was used for mRNA expression in the same tissue of the same isoform at different periods and in the same tissue of different isoforms at the same period. One-way ANOVA was used to analyze the mRNA expression of the same isoform in different tissues at the same period). All calculations were performed with the SAS 9.2 software (SAS Inst., Cary, North Carolina, USA). GraphPad prism 6.0 software was used to make the chart. Differences were considered significant at $P < 0.05$. All data are expressed as the mean ± standard error (S.E.).

## RESULTS

### Analysis of *L-PRLR* and *S-PRLR* and their associated gene sequences

The measured molecular weights of *L-PRLR* and *S-PRLR* are 65,235.36 KD and 33,847.48 KD, respectively, and the isoelectric points are 5.12 and 8.34, respectively.

The nucleotide sequence similarity of the CDS region between ovis and other species of *L-PRLR* and *S-PRLR* are shown in Table 1. The nucleotide sequences of the *L-PRLR* and *S-PRLR* genes are highly conserved among ruminants.

The secondary protein structures of *L-PRLR* and *S-PRLR* are different. The secondary protein structure of *L-PRLR* includes alpha helix, extended strand, random coils and β-turns at proportions of 16.01%, 21%, 59.55% and 3.44%, respectively,whereas the secondary protein structure of *S-PRLR* contains only alpha helices, extended strand and random coils, at proportions of 18.24%, 30.07% and 48.99%, respectively. The tertiary structures of *L-PRLR* and *S-PRLR* encoding proteins are mainly composed of extended fragments and random coils (Fig. 1). The forecast results are consistent with the secondary structures.

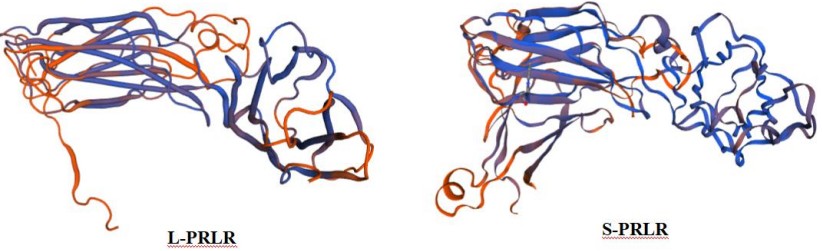

L-PRLR                    S-PRLR

**Figure 1** **Prediction of tertiary structure of L-PRLR and S-PRLR protein in sheep.** Prediction of tertiary structure of L-PRLR and S-PRLR protein in sheep.

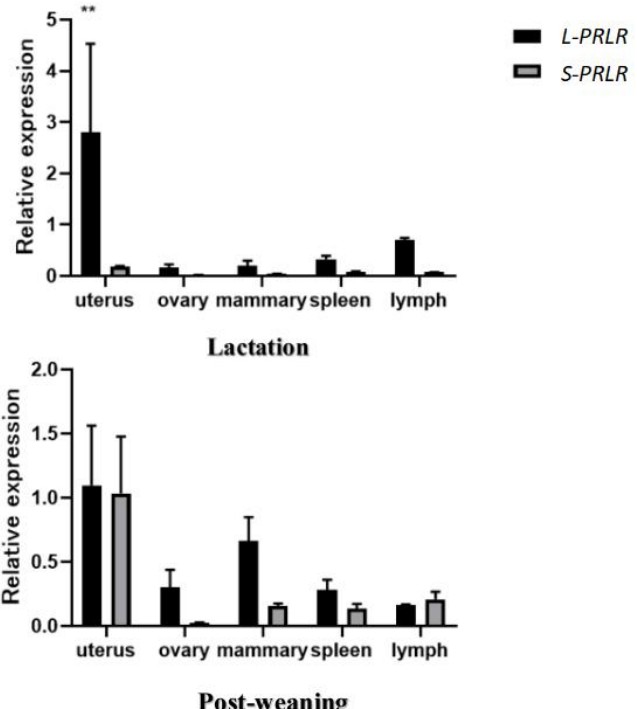

**Figure 2** **MRNA expression of *L-PRLR* and *S-PRLR* in different tissues during lactation and post-weaning in sheep.** Asterisks (**) indicate that the difference is extremely significant ($P < 0.01$).

## mRNA expression of *L-PRLR* and *S-PRLR* in different tissues of sheep

All samples were measured by spectrophotometer and had $OD_{260}/OD_{280}$ values between 1.8 and 2.0, which indicated that they were of good purity.

The mRNA expression of *L-PRLR* and *S-PRLR* in uterus, mammary gland, ovary, spleen and lymph tissue during lactation and post-weaning periods are shown in Fig. 2 and Table 2.

**Table 2  Relative expression of *L-PRLR* and *S-PRLR* in different tissues during lactation and post-weaning in sheep.**

|  |  | Uterus | Ovary | Mammary gland | Spleen | Lymph tissue |
|---|---|---|---|---|---|---|
| *L-PRLR* | Lactation | 2.808 ± 1.725[a] | 0.164 ± 0.064[b] | 0.194 ± 0.102[b] | 0.330 ± 0.065[b] | 0.700 ± 0.045[ab**] |
|  | Post-weaning | 1.098 ± 0.465[a] | 0.305 ± 0.134[ab] | 0.662 ± 0.188[ab] | 0.282 ± 0.079[ab] | 0.160 ± 0.009[b] |
| *S-PRLR* | Lactation | 0.186 ± 0.007 | 0.022 ± 0.002 | 0.038 ± 0.004 | 0.076 ± 0.014 | 0.072 ± 0.002 |
|  | Post-weaning | 1.034 ± 0.444[a**] | 0.022 ± 0.006[b] | 0.156 ± 0.020[ab**] | 0.138 ± 0.035[b] | 0.208 ± 0.060[ab] |

Notes.
Means with different superscripts within the same row are significantly different ($P < 0.05$).
Asterisks (**) within the same column represent significant differences ($P < 0.01$).

The mRNA expression of *L-PRLR* in uterus, mammary gland, ovary, spleen and lymph tissue during lactation and post-weaning periods in ewes are shown in Table 2. During lactation mRNA expression of *L-PRLR* was higher in the uterus than in the ovary, mammary gland and spleen ($P < 0.05$) but was similar to that of lymph tissue. However, during the post-weaning period, mRNA expression of *L-PRLR* in the uterus was higher than that in lymph tissue ($P < 0.05$), but it was not different from the values measured in the uterus, ovary, spleen and mammary gland. The expression in lymph tissue was higher during lactation than in the post-weaning period ($P < 0.01$) but there were no period-related differences in expression in the case of uterus, ovary, mammary gland or spleen.

The mRNA of *S-PRLR* in uterus, mammary gland, ovary, spleen and lymph tissue during lactation and post-weaning periods in ewes are shown in Table 2. There were no differences in mRNA expression of *S-PRLR* among the five tissues during the lactation period. However, during the post-weaning period mRNA expression of *S-PRLR* in uterus was higher than in the ovary and spleen ($P < 0.05$), although it was similar to that of the mammary gland and lymph tissue. No differences were observed in the mRNA expression of *S-PRLR* among ovary, mammary gland, spleen and lymph tissue during post-weaning period. Expression of *S-PRLR* in the uterus and mammary gland was lower during lactation than in the post-weaning period ($P < 0.01$), but there were no period-related differences in the cases of ovary, spleen or lymph tissue.

The expression of *L-PRLR* and *S-PRLR* in 5 tissue during the lactation and post-weaning period are shown in Fig. 2. Expression of *L-PRLR* in the uterus was higher than that of *S-PRLR* during the lactation period ($P < 0.01$). However, there were no differences in expression of the two genes in the ovary, spleen, mammary gland and lymph tissue during the lactation period nor within any of the five tissues during the post-weaning period.

## DISCUSSION

### Sequence analysis of prolactin receptor isoforms

A long form (*L-PRLR*), an intermediate form (*I-PRLR*) and two short forms (*S-PRLR*) occur in humans (*Trott et al., 2003*; *Abramicheva & Smirnova, 2019*). There are at least four *PRLR* isoforms in mice, including three short and one long, but only two of the short *PRLR* isoforms are considered to be proteins (*Tan et al., 2011*) and there are two *PRLR* isoforms (*L-PRLR* and *S-PRLR*) in rats (*Jiang et al., 2004*). Cloning and genomic analysis of cDNA has revealed that *L-PRLR* and *S-PRLR* have arisen from the process of

differential alternative splicing of their coding genes (*Moore & Oka, 1993*). *S-PRLR* differs from *L-PRLR* by having a 39 base pair insert at the beginning of the cytoplasmic domain, but it has two contiguous inframe stop codons at its 3′end (*Bignon et al., 1997*). The present study also reveals differences in the secondary and tertiary structure of the two forms of the receptor protein, which is consistent with there being differences in expression and function of the two protein forms. Our study also showed that the nucleotide sequences of *L-PRLR* and *S-PRLR* genes in the sheep were similar to those of other species, indicating that these are highly conserved among ruminants, and could explain why *L-PRLR* and *S-PRLR* have similar functions across different species.

## The expression of prolactin receptor isoforms in different tissues

*L-PRLR* has been reported as predominantly expressed in the mammary gland, ovary, liver, uterus, skeletal muscle, corpus luteum, and adrenal glands of goats (*Shi et al., 2016*). This is in keeping with the important role of prolactin receptors as the mediators of prolactin's actions in processes such as growth, lactation, reproduction and immunity (*Posner et al., 1974*). Our results confirm the presence of *L-PRLR* and *S-PRLR* in tissue from the uterus, mammary gland, ovary, spleen and a lymph tissue in ewes during lactation and the post-weaning period. However, we have found some differences in mRNA expression of *L-PRLR* and *S-PRLR* throughout these tissues in sheep. For instance, the high level of expression of *PRLR* we recorded in the sheep uterus in both periods that were sampled here accords with a similar finding in black Muscovy ducks (*Li et al., 2020*). Prolactin probably has a vital role in the post-lambing regeneration of the uterine epithelium, which is generally completed within 31 days (*O'Shea & Wright, 1984*). This process involves reduction of the uterine volume, some tissue degradation, and epithelial repair of the endometrium (*Tielgy et al., 1982*). The higher mRNA expression of *L-PRLR* in uterine tissue of the ewes, compared with that of *S-PRLR,* during lactation and post-weaning indicates that the long isoform of the receptor may be primarily involved in uterine repair and recovery following birth of lambs. The increase in expression of *S-PRLR* in the uterus from lactation to post-weaning indicates that the short form of the receptor may have a primary role during normal maintenance of the uterus.

 *L-PRLR* and *S-PRLR* appear to serve different roles in ovary as well. For example, PMSG increased mRNA expression of *L-PRLR*, suggesting a possible involvement of *L-PRLR* in folliculogenesis. In contrast, hCG treatment stimulated expression of *S-PRLR*, indicating a role for the corresponding receptor isoform in formation and maintenance of the corpus luteum (*Thompson et al., 2011*). Also, reverse-transcription PCR analysis of sheep ovarian tissue showed differences in localization and expression of both *S-PRLR* and *L-PRLR* throughout the estrous cycle, with *L-PRLR* being particularly localized in stromal cells surrounding primordial and primary follicles, whereas genes for both *PRLR* isoforms were found in granulosa cells of preantral follicles and luteal cells within the corpus luteum (*Picazo et al., 2004*). The expression of *L-PRLR* in the sheep ovary is markedly increased around the time of estrus, unlike that of *S-PRLR* which does not differ throughout the estrous cycle (*Picazo et al., 2004*). However, during the lactation period follicle development is inhibited and estrus does not occur (*Song et al., 2019*). In the case of *L-PRLR* our results

show an increased level of expression during the post-weaning period which is consistent with a role for the long form of the receptor during the recovery of ovarian follicular development following the birth of lambs.

In goats, expression of *PRLR* gradually increases during the dry period after lactation (*Song et al., 2019*) in a pattern similar to the increases in expression levels of *L-PRLR* and *S-PRLR* recorded in the mammary glands of ewes in the present study. At the end of lactation mammary glands enter a degenerative phase, in which the fat pad regenerates in concert with increases in *PRLR* mRNA content of the adipocytes (*Lesueur et al., 1991*). This finding and the present results implicate involvement of L-PRLR and *S-PRLR* in the process of post-lactational mammary remodeling. *L-PRLR* has a principal role in the induction of milk protein gene transcription (*Das & Vonderhaar, 1995*) whereas the involvement of *S-PRLR* is less clear. For instance, *L-PRLR* could activate the β-casein gene promoter, while *S-PRLR* did not (*Berlanga et al., 1997*). Also, over-expression of *S-PRLR* enabled mammary development and function when *PRLR* genes were knocked out in heterozygous mice (*Zi, Chen & Wang, 2012*) and *S-PRLR* appears to have a negative role in relation to the involvement of PRL in milk protein gene transcription (*Berlanga et al., 1997*). Together with the present finding of higher expression of *L-PRLR* compared with that of *S-PRLR* in the ovine mammary gland, this is evidence for *L-PRLR* having the predominant role on the maintenance of lactation and in mammary gland repair processes.

PRL play an crucial role in regulating immunity (*Gharbaran et al., 2020*), not only by promoting the proliferation of immune cells, but also by stimulating the production of antibodies such as IgG and IgM, and these functions involve expression of the PRL receptor genes (*Zhou et al., 2020*). For instance, the transcripts encoding both isoforms of PRLR (*L-PRLR* and *S-PRLR*) were recorded in all lymphoid tissues examined in mice and rats (*Touraine & Kelly, 1995*) and in the spleens and lymph tissues of sheep in our study. In the case of thymus and spleen, mRNA expression of *L-PRLR* was higher than that of *S-PRLR* (*Ouhtit, Kelly & Morel, 1994*). In the present study, expression of *L-PRLR* was higher than that of *S-PRLR* in lymphatic tissue during lactation, but this pattern reversed during the post-weaning period. These differential findings indicate different specific roles for the two isoforms of the receptor in relation to their involvement in the progression of immune responses during and after lactation that warrant further study.

## CONCLUSIONS

It was concluded that the mRNA expression of *L-PRLR* and *S-PRLR* genes varies among different tissues in sheep. The mRNA expression of *L-PRLR* and *S-PRLR* is higher in the uterus than in the ovary, spleen, mammary gland and lymph tissue during both lactation and the post-weaning period. The mRNA expression of *L-PRLR* is higher than *S-PRLR* in the uterus, ovary, spleen, mammary gland during both lactation and post-weaning period. The mRNA expression of *L-PRLR* was higher than *S-PRLR* in lymph tissue during lactation, but this pattern reversed during the post-weaning period.

## ACKNOWLEDGEMENTS

The authors express their gratitude to Doctor Zhang for laboratory support and to fellow students in the laboratory for assistance.

### Funding

This research was supported by the National Modern Agricultural Industry Technology System Construction Project of China [CARS-38] and [CARS-39], the Excellent Youth Program of Hebei Province [8042018-1081034], and the Hebei Province Science Foundation for Youths (C2019204357). The funders had no role in study design, data collection and analysis, decision to publish, or preparation of the manuscript.

### Grant Disclosures

The following grant information was disclosed by the authors:
National Modern Agricultural Industry Technology System Construction Project of China: CARS-38, CARS-39.
Excellent Youth Program of Hebei Province: 8042018-1081034.
Hebei Province Science Foundation for Youths: C2019204357.

### Competing Interests

The authors declare there are no competing interests.

### Author Contributions

- Ruochen Yang and Chunhui Duan conceived and designed the experiments, performed the experiments, analyzed the data, prepared figures and/or tables, authored or reviewed drafts of the paper, and approved the final draft.
- Yunxia Guo conceived and designed the experiments, analyzed the data, prepared figures and/or tables, and approved the final draft.
- Yujing Ma analyzed the data, prepared figures and/or tables, and approved the final draft.
- Nazi Niu performed the experiments, analyzed the data, prepared figures and/or tables, and approved the final draft.
- Yingjie Zhang conceived and designed the experiments, analyzed the data, authored or reviewed drafts of the paper, and approved the final draft.
- Yueqin Liu conceived and designed the experiments, authored or reviewed drafts of the paper, and approved the final draft.

### Animal Ethics

The following information was supplied relating to ethical approvals (i.e., approving body and any reference numbers):

Hebei, P.R. China; permit number 2018082

## DNA Deposition

The following information was supplied regarding the deposition of DNA sequences:

The Prolactin receptor data is available at UniProt: O46561

- L-PRLR(Ovis): identifier: O46561-1
- S-PRLR(Ovis): identifier: O46561-2

The Ovis aries glyceraldehyde-3-phosphate dehydrogenase (GAPDH), mRNA sequence is available at NCBI: NM_001190390.1.

## Data Availability

The raw measurements are available in the Supplemental File.

## Supplemental Information

Supplemental information for this article can be found online at http://dx.doi.org/10.7717/peerj.11868#supplemental-information.

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
