# Peer review of "Sequence analysis and mRNA expression of prolactin receptor gene isoforms in different tissues of sheep during lactation and the post-weaning period"

_PeerJ, doi:10.7717/peerj.11868_

## Round 0.1 · original submission · Major Revisions

The manuscript by Ruochen Yang et al has been written in a casual way without a particular hypothesis. The design of the study is not well focused. If the manuscript is about the study of the expression of L-PRLR and S-PRLR in different tissues of lactating sheep; what was the purpose to analyze the protein structure of the said genes.? How the structural analysis and phylogenetic tree can address the questions of prolactin involvement during lactation and the post-weaning period of sheep? The structural analysis results are superfluous. In fact, the protein expression by western blotting of prolactin genes would have been meaningful. The methodology for real-time expression is confusing with respect to primer details and reaction setup etc. The information about sequencing details of genes and their submission to NCBI etc. is missing. The results have not been well written and the discussion of the study lacks connectivity. The conclusion made by the authors is inappropriate.

If the manuscript in the second revision is not upgraded in light of the above observation, I will not consider it further.

Reviewer 1 ·

Basic reporting

This is a well-written and properly focused article according to the scope of the Journal.
It includes sufficient introduction material and appropriate supporting references.
The article shows the structure of a scientific manuscript, and it includes descriptive and informative tables and figures.
However, references appear to be incorrectly spelled. Please follow guidelines described in the "Reference Format" section of the Journal.

Experimental design

The design of this research seems to be original; however, neither the hypothesis nor the research question are clearly described in the article.
The experimental units (i.e., ewes) are well described, only that it is not clear why the food consumption was measured.
Laboratory and bioinformatics procedures are well described.
The statistical analyses section seems to be insufficient to support and validate the results obtained. Specifically, the statistical model is missing. According to the data analyzed, the statistical model should include at least three main effects: PRLR isoform, physiological state and sampled tissue.
In addition, it is important to clarify which was the statistical test used to compare mean values for mRNA expression among the different tissues (i.e., Duncan test, Turkey test, etc.).
The statistical support of the experiment should be clearer and more complete so that the study can be replicated.

Validity of the findings

Discussion section is well focused and provided the explanation of the results reported; also this section has been strengthened due to the inclusion of several good supporting references.
However, the missing part is the discussion according to the effect of the physiological state of the ewes (i.e., lactation and post-weaning) on the mRNA expression. Such results were reported but not discussed.

Additional comments

- Abstract section does not include information or results from the phylogenetic tree obtained.
- Lines 39-42 According to Table 2, it seems to be that S-PRLR similarity between Ovis aries and Mus musculus is very low, so it does not agree with the statement described in these lines of the article.
- Lines 127-128 What about Mus muscles sequence for S-PRLR?
- Lines 138-139 This information corresponds to the Materials and Methods section.
- When a statistical comparison is not significant, the P-value is generally not needed (i.e., P>0.05).
- Results described by Figure 4 are also included in Table 3. I suggest to clarify which effects are studied in Table 3 and which in Figure 4.

Reviewer 2 ·

Basic reporting

1. Based on paper title, there was no summary of cDNA sequence of both isoforms of gene.
2. In table 2 they mentioned CDS similarity with other species,but the alignment files, accession number of sequenced cDNA not mentioned in text.
3. References of the paper is not uniform. The all references should be arranged in Journal pattern.
4. Kindly concise article discussion with our related studies, need not be given other important role played by gene. It should be elaborated in introduction section. That why you planned your study.
5. Kindly improve the article language and concise the discussed text.
6. According to my view it should be published as Short note after following revision.

Experimental design

Experimental design of the present study is good.

Validity of the findings

Validity of the present study is as generally reported by earlier worker. It is general type study.

Additional comments

Kindly rewrite the article according to journal short not pattern. The discussion should be restricted to their study only. The References given in text is also minimized.

---

## Round 0.2 · Major Revisions

The manuscript by Yang et al is still full of flaws and not improved much in light of the earlier comments. As pointed out by one of the reviewers the manuscript is a very basic work studying the expression PRLR (L-PRLR) and short PRLR (S-PRLR) by Real-time PCR in different tissues of sheep during lactation and post-weaning. In the present era of next-generation sequencing, the manuscript does not offer a clear contribution to the literature. Hence, the manuscript is not recommended for publication in PeerJ unless you can address the remaining reviewer comments.

Reviewer 1 ·

Basic reporting

Thank you to the authors for their kind response to all comments. I consider that the manuscript has improved significantly.
However, I still have two comments:
1) I suggest that Conclusions section could be improved if the authors include some information about main differences in RNA expression between isoforms, between physiological periods, as well as among tissues.
2) I suggest that References section should be revised carefully. It appears to be that "name of authors" and "journal name" need to be corrected according to the "guide for authors".

Experimental design

No comment.

Validity of the findings

No comment.

Additional comments

Thank you to the authors for their kind response to all comments. I consider that the manuscript has improved significantly.
However, I still have two comments:
1) I suggest that Conclusions section could be improved if the authors include some information about main differences in RNA expression between isoforms, between physiological periods, as well as among tissues.
2) I suggest that References section should be revised carefully. It appears to be that "name of authors" and "journal name" need to be corrected according to the "guide for authors".

·

Basic reporting

The authors performed experiments in sheep to mRNA expression of prolactin receptor gene isoforms in different tissue of sheep during lactation and post-weaning (Fig 3/Table 3).

The authors also did the bioinformatic analysis using the published sequences of the prolactin receptor gene isoforms to predict the nucleotide similarity of the gene (Table Table 2), and secondary protein structure of the isoforms (Figure 1).


The only contribution of the manuscript is the mRNA expression of the prolactin receptors isoforms in different tissue



A) The language of the article needs improvement. There are sentences with inexact expressions. Below are a few examples:-

1) Abstract section Line 52.... The word"probably" shows the uncertainty of the results and needs to be stated exactly.
2.) Introduction section Line 72..." would of various in different tissues of sheep"
3) Result section Line 151, 161, 170 and many other places...." 5 times"
4) Result section Line 158... "Lymph tissue" instead of the lymph node.


B) There is incoherence in the structure of the abstract, introduction, and discussion part of the article
1) Abstract: The large portion of the abstract is dedicated to research methodology and does not provide background information of the research, a statement on the research problem, the rationale/ goal of the research, and significant and/or implication of the research finding.
2) Introduction: The introduction is deficient in the background information on the research problem, does not elaborate on the role of PRLR in reproduction/ immunity, and does not elaborate on the rationale of the research. Some of the discussion portions can be used in the introduction.
3) The discussion is divided into sections and each section seems like a review of the literature. The purpose of the discussion is to interpret and describe the significance of the results in light of what is already known and to explain any new understanding or insights that emerged as a result of the study. Its sectioning needs to be avoided and need to be concise.
4) The conclusion section is ambiguous and does not conform to standards. It should summarise the main findings of the study and should state the significance of the results.

C) Table and figures:
1) Table 1 can be omitted. The sequence of the primers can be written in text rather than put into a separate table
2) Figure 2 is not a result and should be omitted.
3) Figure 3 and Table 3 present the same data. There is no need to put data both in figure and table form. Table 3 should be omitted.

Experimental design

No comment

Validity of the findings

The conclusion is not clearly stated. It gives a general statement. It should be specific. It should summarise the main findings of the study. It should state the significance of the results.

Additional comments

No comment

---

## Round 0.3 · accepted · Accept

All the queries raised by reviewers were properly addressed in the revised manuscript. The manuscript has been significantly improved. I recommend the manuscript for publication.